# Bioconversion of a Dairy By-Product (*Scotta*) into Mannitol-Stabilized Violacein via *Janthinobacterium lividum* Fermentation

**DOI:** 10.3390/microorganisms13092125

**Published:** 2025-09-11

**Authors:** Mario Trupo, Rosaria Alessandra Magarelli, Salvatore Palazzo, Vincenzo Larocca, Maria Martino, Anna Spagnoletta, Alfredo Ambrico

**Affiliations:** Department for Sustainability, ENEA—Italian National Agency for New Technologies, Energy and Sustainable Economic Development, Trisaia Research Center, 75026 Rotondella, Italy; mario.trupo@enea.it (M.T.); rosaria.magarelli@enea.it (R.A.M.); salvatore.palazzo@enea.it (S.P.); vincenzo.larocca@enea.it (V.L.); maria.martino@enea.it (M.M.); anna.spagnoletta@enea.it (A.S.)

**Keywords:** violacein, antibacterial activity, *Janthinobacterium lividum*, dairy by-products, microbial pigment production, mannitol encapsulation

## Abstract

Violacein is a natural pigment with a wide range of biological activities, including antimicrobial, antitumor, and immunostimulatory properties. However, its industrial-scale production is hindered by low yields from microbial fermentation. This study investigated the use of *scotta*, a low-value by-product of the dairy industry, as an alternative and cost-effective substrate for violacein biosynthesis using *Janthinobacterium lividum* DSM1522. Different types of *scotta*, including one derived from lactose-free cheese production, were characterized and tested in flask cultures and a 2 L bioreactor. The results demonstrated that both medium dilution and increased oxygen-transfer coefficient (k_L_a) significantly enhanced violacein production. In the bioreactor, a final yield of 58.72 mg of violacein for each litre of diluted *scotta* was achieved. The pigment was then stabilized through a spray-drying process using mannitol as a carrier, resulting in a water-soluble powder that retained antibacterial activity against *Bacillus subtilis*. The drying process also improved pigment solubility in water, suggesting its potential application in formulations to control Gram-positive bacteria. Overall, this study highlights the potential of *scotta* as a sustainable fermentation substrate and presents a promising encapsulation approach for violacein stabilization. However, further investigations are needed to optimize the spray-drying process, specifically, to characterize the microgranules and to determine their storage stability.

## 1. Introduction

Violacein is a natural pigment with a broad spectrum of biological activities including antimicrobial, antiparasitic, antiprotozoal, anti-inflammatory, antitumor, antileukemic, antioxidant, antiulcerogenic and immunostimulatory effects [1,2,3,4,5,6]. This wide range of biological activities has aroused great attention and has attracted interest in finding a potential application as a therapeutic agent [7].

The IUPAC name of violacein is 3-[2-hydroxy-5-(5-hydroxy-1H-indol-3-yl)-1H-pyrrol-3-yl]indol-2-one, while its chemical formula and molecular weight are C_20_H_13_N_3_O_3_ and 343.3 g mol^−1^, respectively. It belongs to the class of hydroxyindoles and its chemical structure is described in Figure 1 [8].

This compound is synthetized by a limited Gram-negative bacteria species and represents a secondary metabolite associated with biofilm production that can provide a survival advantage for its producing microorganisms in natural environments [9,10,11].

Violacein is a bisindole compound derived by the condensation of two molecules of L-tryptophan, an essential amino acid with an indole ring, which is a precursor of a large number of secondary metabolic pathways in microorganisms [12].

Due to the different biological properties of violacein, it is conceivable that there will potentially be a commercial use in the future, so the development of its large-scale production from microbial synthesis is of great interest [13]. In line with this, *Chromobacterium violaceum* and *Janthinobacterium lividum* produce a well-known violet pigment that is recognized as violacein [14,15]. In some reports, in addition to the need to select producer strains, the composition of the growing medium and the manner in which cultivation conditions can affect the yield of violacein have been evaluated and emphasized [16,17].

Although violacein may have high economic potential, the low yield of violacein through bacterial growth may remain a huge limitation for its application. Various strategies have been applied in seeking to avoid this issue and reduce the production cost of violacein. Researchers are paying particular attention to the identification of low-cost substrates deriving from agro-industrial waste or by-products that have high content levels of the proteins, carbohydrates, minerals and vitamins optimal for the growth of violacein-producing bacteria [18]. In fact, in general, the production of high-value compounds based on microbial growth processes and with the use of conventional media is very expensive [19]. Some works have highlighted the production of violacein by *Chromobacterium violaceum* in different agricultural waste materials, such as sugarcane bagasse, pineapple waste, molasses, brown sugar and agro-industrial waste soybean meal [20,21]. This strategy is also encouraged by the adoption of circular-economy measures to increase the sustainability of agro-industrial processes, with the aim of converting wastes into resources [22]. In fact, the suitable use of food waste and by-products as raw materials could generate economic benefits for the industry, avoiding significant losses of energy, nutrients and economic resources, and generating a sustainable product with a low environmental impact [23].

In this context, it is possible to hypothesize that whey from the dairy-processing industry could be one such source, and one which may be utilized in fermentation systems for violacein biosynthesis, since it is rich in lactose and proteins [24,25].

Whey is the major by-product from the dairy industry and about 50% of the milk produced in Europe in 2020 was used to make cheese, producing 54.8 million tonnes of whey [26]. Generally, the cheese whey is valorised in food-based or pharmaceutical applications as a source of high-added-value products, mainly whey protein isolate, or is utilized as animal feed [27,28]. However, due to the lack of infrastructure for whey processing, especially in small dairies, there is still a high proportion of this whey that is discarded and could create potential environmental problems [29].

In Italy, whey is used in the production of whey-based cheeses called Ricotta. The liquid that remains after this last separation is called second cheese whey or *scotta*, and typically represents more than 90% of the original whey [30], which may be from different milk types (sheep, goat, cow, or buffalo). *Scotta* is a low-value by-product of a low-yield process but holds significant potential due to its residual nutritional content and fermentable sugars. Its current uses are modest, but research and innovation could transform it into valuable resources in the bioenergy, feed, or health-oriented industries [31]. In particular, it could be used as biotechnological substrate for microbial growth to obtain biofuels, bioplastics and food additives such as lactic acid, extracellular polymeric substances (EPS), ethyl lactate, etc. [32].

*Scotta* is generally composed of lactose (about 5.0%), salts, short peptides and some residual whey proteins (about 0.2%), which may still contain essential amino acids as tryptophan [33]. This amino acid is a versatile metabolite that is shunted into several microbial secondary pathways and, in particular, upon which violacein production is directly dependent [4]. L-tryptophan is the sole precursor in the biosynthesis of violacein by five enzymes encoded by the *vioABCDE* operon, and all atoms in the violacein molecule are derived directly from two molecules of tryptophan [12,34].

However, to the best of our knowledge, there is insufficient data regarding the use of these dairy industry by-products as low-cost media to grow bacteria able to produce violet pigments such as violacein.

Therefore, the aim of this study was to find a way to reduce the production cost of violacein. Specifically, we have carried out trials on a well-characterized bacterial strain of *J. lividum* (DSM1522), using a 2 L bench scale fermenter to evaluate the growth and the capability of violacein biosynthesis on *scotta* from Italian dairies, identifying the best growth parameters and evaluating spray drying as a method used to stabilize the violacein extract.

## 2. Materials and Methods

### 2.1. Bacterial Strains

*J. lividum* strain DSM1522 was provided in lyophilized form by the Leibniz Institute DSMZ (Braunschweig, Germany). For reactivation, freeze-dried bacterial cells were rehydrated and grown in nutrient broth (Peptone 5.0 g L^−1^; Meat extract 3.0 g L^−1^).

*Bacillus subtilis* strain ET-1, isolated from soil, was provided in fresh form by ENEA microbial culture collection (EMCC).

The pure cultures were maintained on nutrient agar (NA) plates at 4 °C and images of bacteria cells were acquired, see Figure 2, by optical microscope (Olympus, model BX60, Tokyo, Japan). Furthermore, for the strain DSM1522, the biochemical profile was acquired using Microbial Identification Software version 6.3.1 (Biolog, Inc., Hayward, CA, USA).

### 2.2. Physical–Chemical Characterization of Dairy By-Products (Scotta)

Four different cow’s milk by-products were supplied by small Italian industrial dairy companies and analysed for density, electrical conductivity (EC), pH and lactose content.

Specifically, density was determined using a Quevenne glass lacto-densimeter; pH and electrical conductivity were measured by dipping the electrodes of a calibrated pH-meter (Hanna Instruments, Halo^®^ Wireless pH Meters, Villafranca Padovana, Italy) and a conductivity-meter (WTW, inoLab^®^ Cond Level 1, Washington, DC, USA) into the samples. Lactose determination was performed with an Agilent HPLC system (see Section 2.7).

### 2.3. Shaking-Flask Cultivation

#### 2.3.1. Evaluation of the Effects of Different Temperatures on Violet Pigmentation, Using Different Scotta from Ricotta-Cheese Production

Cultivation trials were carried out in 500 mL shaking flasks severally filled with 50 mL of three different dairy by-products, named *scotta* 1 (S1), *scotta* 2 (S2) and *scotta* 3 (S3).

Before sterilization, the pH was adjusted to approximately 7.0 by adding 1 M NaOH. All flasks were inoculated with 500 µL of bacterial suspension (approximately 1.0 × 10^8^ CFU mL^−1^) obtained by diluting bacterial colonies cultured on NA for 96 h. Then, the flasks were incubated at 18, 22 and 28 °C in a thermostatic orbital shaker at 130 rpm. After 120 h, 5 mL samples were centrifuged at 8000× *g* for 10 min, and then the supernatant was discarded, while the pellet with the bacterial cells was resuspended in 1 mL of ethanol (99%) and vortexed for 5 min.

Finally, the ethanolic suspension was re-centrifuged under the same conditions and the supernatant was used for spectrophotometric analyses of violacein. The experiments were conducted in triplicate and repeated twice.

#### 2.3.2. Evaluation of Different Volumetric Mass Transfer Coefficients (k_L_a) and the Effect of Scotta Dilution on Violacein Production

To evaluate the effect of dilution on violacein production, 500 mL Erlenmeyer flasks were filled with 50 mL samples of diluted and undiluted S3, inoculated with 500 µL of bacterial suspension, and incubated at 22 °C. Since, as reported in the appropriate paragraph, S1 provided the best results in terms of violacein production, this dilution factor was chosen in order to obtain conditions like those observed in S1, in terms of lactose concentration and EC value (approximately 10 g L^−1^ and 5000 µS, respectively). To obtain these values, 18.5 mL of S3 was diluted with 31.5 mL of deionized water.

Furthermore, we evaluated how the different volumetric mass transfer coefficients (k_L_a) of 135 h^−1^ and 34 h^−1^ influenced violacein production. For this purpose, to achieve these two different k_L_a values, 500 mL Erlenmeyer flasks with and without baffles were used under the same culture conditions (50 mL of culture medium and 130 rpm).

The k_L_a values were determined using a k_L_a calculator [35], and based on measurements with PreSens’ SFS v4 plastic shake flasks at 50 mm shaking diameter and a PBS buffer in the temperature range of 30–37 °C. The website calculator, based on some data input information (flask volume, culture medium volume, and rotation speed), estimates the k_L_a values for other diameters according to Equation (1):(1)kLaother = kLa50 mm × dother 50 mm0.33

After 120 h, 5 mL samples were collected and analysed as above to quantify the violacein production.

These two experiments were carried out in triplicate and repeated two times.

#### 2.3.3. Indirect Impact of Lactose Hydrolysis on Pigmentation

From the biochemical profile, as shown in Appendix A, it is possible to note that *J. lividum* strain DSM 1522 is a lactose-negative bacterium, although it utilizes the hydrolysis products glucose and galactose. Therefore, the effect of lactose hydrolysis on violacein biosynthesis was evaluated by comparing samples of S3 with and without the addition of a lactase enzyme.

Specifically, following the indication reported on the data sheet of the Delact Plus enzyme supplied by Alce International s.r.l. (Quistello, Italy), 1 mL of enzyme was added aseptically to 1 L of S3, using 0.22 μm syringe filters.

Furthermore, trials were also planned using S4, a dairy by-product derived from the industrial production of lactose-free cheeses.

The experimental design included 6 sample types, and the trials were carried out in 500 mL Erlenmeyer flasks filled with 50 mL of S3, S3+L (S3 with lactase added) and S4, diluted or undiluted, maintaining a k_L_a of 135 h^−1^.

The same dilution factor, as specified in the previous paragraph, was used for all sample types. Sample type S4, although already presenting a low lactose concentration at the start, was diluted using the same dilution factor used for S3. This is because the purpose of dilution was to reduce the concentrations of all components present in the serum, including glucose and galactose.

After 120 h, 5 mL samples were centrifuged at 8000× *g* for 10 min. The bacterial cell pellet was then used to determine the level of violacein, while the supernatant was used to quantify sugars by HPLC analysis. The experiments were conducted in triplicate and repeated twice.

### 2.4. Violacein Production in a Bioreactor at Two Different k_L_a

Based on the tests conducted in Section 2.3, considering the indirect impact of lactose hydrolysis on pigmentation, the production of violacein was evaluated in a 2 L bioreactor (BIOSTAT B, Braun Biotech International, Melsungen, Germany) using diluted lactose-free *scotta* S4. The experiments were conducted to verify the crucial role of oxygen transfer in violacein production. For this purpose, to avoid technical problems due to the high-velocity stirring, the trials were not conducted at the same k_L_a value used in the flasks (135 h^−1^). In fact, based on preliminary tests, at a stirring of over 720 rpm and an air flow of 0.9 Lpm, an excessive foaming formation was observed.

The bioreactor used was a 13 cm diameter stirring vessel equipped with two Rushton impellers (4 cm diameter) with six blades equally spaced around the disc of the impeller and a sparger ring located under the lower impeller. The culture was carried out in batch mode with a working volume of 1.2 L. A preculture aged 96 h was used as starter at a rate of 10% (*v*/*v*).

During the fermentation process, the liquid culture temperature was maintained at 22 °C and a pH of 7.2 by adding 1 M sodium hydroxide solution and 1 M hydrochloric acid solution. To achieve the k_L_a values of 14.97 and 58.72 h^−1^, the aeration was maintained at 0.2 and 0.9 Lpm and the agitation was maintained at 400 and 720 rpm, respectively. To predict the k_L_a for this bioreactor system at the two different aeration and agitation rates, the following empirical equation, Equation (2), as proposed by Riet (1979) [36] was used.(2)kLa=C×PgVα×νsβ
where
k_L_a = volumetric liquid-side mass transfer coefficient (h^−1^);*Pg* = total impeller power consumption (W);*V* = liquid volume in vessel (m^3^);*ν_s_* = superficial gas velocity (m h^−1^).*C*, *α*, *β* are empirical constants, and depend on the geometrical parameters of the vessel; they are 0.026, 0.4 and 0.5, respectively.

Samples were aseptically recovered from the bioreactor every 24 h and used to quantify the sugars and violacein concentration. Additionally, an aliquot of samples was used to estimate viable bacterial cells by plating sample dilutions on nutrient agar. The bacterial fermentations in the bioreactor were repeated three times. At the end of the fermentation processes, the liquid culture was recovered, and the bacterial cells, separated by centrifugation, were used for pigment extraction.

### 2.5. Extraction and Spray-Drying of Violet Pigment

After the fermentation in the bioreactor, the pigment of the violacein was extracted from the bacterial cells and subsequently stabilized by the spray-drying technique, adopting, as partially modified, the method reported by Venil et al. (2015) [37].

Specifically, bacterial biomass from 1 L of culture was recovered by centrifugation (Avanti J-25 Beckman Coulter, Brea, CA, USA) at 8000× *g* for 10 min, resuspended in 0.2 L of ethanol (99%) and mixed in an orbital shaker at 110 rpm and at room temperature for 30 min.

After this time, the bacterial suspension was centrifuged again and the decolorized biomass was discarded, while 0.2 L of concentrated pigment was added to 0.6 L of mannitol (Biolife Italiana srl, 41BET0192, Milano, Italy) water solution (10%) (*w*/*v*), used as carrier agent. The mixture was homogenized in a magnetic stirrer, kept at room temperature and spray dried.

The mixture was introduced into a spray dryer (APT-2.0 Spray Dryer, Novara, Italy) via a peristaltic pump at 17 rpm, using a silicone tube with an internal diameter of 0.2 mm. Atomization was carried out using a single nozzle (0.8 mm) and the air flow rate at the nozzle was kept constant at 1.6 m^3^ h^−1^.

The inlet temperature was set at 180 °C, while the outlet temperature was approximately 80 °C. Before and after the process performed under these operating conditions, the spray dryer was operated with water for at least 15 min. At the end of the spray-drying process, the powder was collected, weighed and packaged under nitrogen in crown-capped bottles. The bottles were stored at room temperature until being used for antimicrobial testing.

### 2.6. Antimicrobial Activity of Spray-Dried Pigment Against Bacillus Subtilis

The antimicrobial activity of the spray-dried pigment was tested against the Gram-positive bacterium (*Bacillus subtilis* strain ET-1) using the agar well-diffusion test. Specifically, nutrient agar plates were inoculated by spreading 100 µL of bacterial suspension (approximately 1.0 × 10^6^ CFU mL^−1^) from overnight cultures grown at 26 °C on a nutrient broth. Then, 8 mm diameter wells were created on the agar surface using the back of a sterile Pasteur pipet.

Before testing antibacterial activities, 150 mg of encapsuled sample was dissolved in 1 mL of sterile water and the following stock solutions were prepared: 0.30 mg mL^−1^, 3.00 mg mL^−1^, 9.00 mg mL^−1^, 9.37 mg mL^−1^, 18.75 mg mL^−1^, 37.5 mg mL^−1^, 75 mg mL^−1^ and 150 mg mL^−1^.

The wells were filled with 50 μL of each solution, at different pigment concentrations.

In addition, the same aliquots of ethanol and crude extract (not spray-dried) were also used as control experiments. The plates were incubated at 25 °C, and after 96 h the presence of an inhibition zone was determined to check for antimicrobial activity. The experiment was carried out in triplicate and repeated three times.

### 2.7. Sugars HPLC Determination

For the quantification of sugars, an Agilent 1200 series HPLC system (Agilent Technologies, Santa Clara, CA, USA) consisting of an in-line degasser (G1379B), binary pump (G1312B), auto-sampler (G1367B), column temperature controller (G1316A), UV-Vis detector (G1314B), Diode Array detector (DAD) (G1315A) and Refractive Index Detector (RID) was used.

Prior to analysis, 1 mL of sample was centrifuged at 13,000× *g* for 2 min. Then, supernatant was filtered through a 0.22 µm Millipore filter and added to 2 mL vials. The analyses were performed with an Agilent Hi-Plex H analytical column (7.7 × 300 mm, 8 μm), using 0.005 M sulphuric acid solution as the mobile phase under isocratic conditions at a flow rate of 0.7 mL min^−1^. The injection volume was 20 μL and the column temperature was maintained at 60 °C.

The HPLC peaks of the samples were detected by RID and were identified by comparing their retention times with those of external standards of glucose, galactose and lactose. For each standard, the calibration curve was calculated on at least 3 concentrations (between 0.05 and 10 g L^−1^), and each showed good linearity (R^2^ ≥ 0.9999). All data were collected and analysed using OpenLAB CDS Chemstation Edition Rev. C.01.10(201) software. The analyses were performed in three replicates.

### 2.8. Pigment Determination

The crude extract samples were subjected to reverse-phase HPLC using a Zorbax Rx-C18 (Santa Clara, CA, USA) analytical column (4.6 × 250 mm, 5 µm) and analysed to evaluate the presence of violacein. To this end, 20 μL of sample were eluted with 50% isocratic acetone at 0.4 mL min^−1^ and the column temperature was kept at 25 °C. Direct UV absorption detection was performed at the wavelength characteristic of violacein (580 nm) and online spectra between 210 and 720 nm were recorded by DAD.

The HPLC chromatograms and UV-Vis spectra were compared (see Figure 3) with those of an external violacein standard purchased from Merk (Sigma-Aldrich, V9389, Burlington, MA, USA).

During fermentation, the estimate of the amount of pigment was performed by an optical density method, using a spectrophotometer (Thermo Scientific-Multiscan GO, Waltham, MA, USA) at 580 nm. Specifically, the violacein standard was used to prepare at least 4 concentration points (between 0.1 and 0.0125 g L^−1^) to calculate a calibration curve (see Appendix A) and use the resulting equation (y = 29.32x − 0.004) to convert the crude extract sample readings to violacein concentration levels.

The violacein yield (*Vy*) was expressed as g L^−1^ and calculated using Equation (3):(3)Vy=Cve×VsVbc
where *Cve* is the violacein concentration in ethanol, *Vs* is the volume of solvent (ethanol) and *Vbc* is the volume of bacterial culture.

## 3. Results and Discussion

### 3.1. Physico-Chemical Characterization of Dairy By-Products (Scotta)

Dairy industry by-products have been considered as low-cost materials which could be evaluated as alternative substrates in microbial production processes [38].

In Italy, whey is used to produce Ricotta cheese by heating it to approximately 90 °C, and the main by-product of the Ricotta production process is *scotta* [39].

In this study, four different types of *scotta* derived from cow’s milk were supplied by dairy factories located in Lucania, a region of Southern Italy. The physico-chemical characterization of the *scotta* was performed regarding lactose content, pH, density and EC. The data, presented in Table 1, showed significant differences between the four *scotta*, likely due to differences in the initial whey content resulting from the manufacture of different types of dairy products.

A large variability was observed in the lactose content, which, however, confirms that *scotta* was a potential substrate for fermentations [31]. Except for S4, which comes from the production of low-lactose products and contained approximately 7.22 g L^−1^, the lowest lactose level was determined to be that of S1, with 9.99 g L^−1^, while for S2 and S3 42.19 and 26.81 g L^−1^ were recorded, respectively.

As regards pH and EC, samples S1, S2 and S4 did not present particularly different values, ranging between 6.77 and 7.07 and between 5.15 and 7.21 mS cm^−1^, respectively. For S3, a pH value of 3.9 and an EC value of 19.79 mS cm^−1^ were recorded. These higher values for acidity and conductivity could be related to bacterial fermentation or to the use of organic acids, (between 1.5 and 2.5%) and salt in Ricotta manufacture, which, in some cases, may be added to promote the whey protein agglomeration [40,41].

Regarding the density, the values were in a range from 1.007 g mL^−1^ (S1) to 1.027 g mL^−1^ (S4); despite the fact that the differences were small, they reflected the variations in the dissolved solids.

Concerning S1, the set of characteristics reported in Table 1 suggests that the sample may have been diluted during the manufacturing process. In particular, the low level of lactose detected did not appear to be associated with bacterial fermentation, as no decrease in pH or increase in electrical conductivity due to the production of acids was observed.

### 3.2. Shaking-Flask Cultivations

#### 3.2.1. Evaluation of Differences in Temperature on Violet Pigmentation, Using Different Scotta from Ricotta-Cheese Production

Regarding the production of violacein at different temperatures for the three scotta samples (S1, S2 and S3), the data in Figure 4 report that at 18 °C, S1 shows the highest production of violacein (20.91 mg L^−1^), while S2 and S3 show much lower values (3.72 and 1.98 mg L^−1^, respectively).

At 22 °C, S1 still shows the highest production of violacein (21.88 mg L^−1^), while S2 and S3 show increasing production compared to 18 °C (5.24 and 3.76 mg L^−1^, respectively). Finally, at 28 °C, S1 shows a decrease in violacein production (18.81 mg L^−1^), while S2 and S3 present lower production values compared to 22 °C (2.89 and 2.18 mg L^−1^, respectively).

The results are in agreement with those published by Baricz (2018) [16], who determined the best violacein production at 22 °C for *J. lividum* (strain ROICE173) using a synthetic medium R2 (composed, in g L^−1^, of casein 0.25; peptone 0.25; casaminic acids 0.5; yeast extract 0.5; dextrose 0.5; starch 0.5; dipotassium phosphate 0.3; magnesium sulphate 7H_2_O 0.05; and sodium pyruvate 0.3). However, these authors obtained lower yields than us, with violacein concentrations of 0.01 and 0.13 mg L^−1^ at 30 and 22 °C, respectively.

Considering these data, we can state that *J. lividum* DSM1522 synthesized the highest amount of violacein in S1 medium at all temperatures, although slightly higher values were observed at 22 °C. This trend is also observed for S2 and S3, which both show a lower overall production of violacein, with a notable increase in production at 22 °C compared to 18 °C and 28 °C.

These results lead us to hypothesize that the increased violacein synthesis observed in substrate S1 could be due to the potential dilution of the *scotta* during the production process. Indeed, a combination of factors such as lower nutrient concentration, favourable density, and moderate electrical conductivity may induce metabolic stress that can lead to increased violacein synthesis in diluted *scotta* compared to other samples.

It is already known that abiotic stress could play a key role in the activation of the violacein biosynthetic pathway. In fact, violacein acts as a protective mechanism against microbial competitors and oxidative damage, and functions as part of biofilm formation [9].

#### 3.2.2. Evaluation of Different Volumetric Mass Transfer Coefficients (k_L_a) and the Effect of Scotta Dilution on Violacein Production

In this study, the effects of oxygen-transfer rate (expressed in k_L_a) and *scotta* dilution on violacein biosynthesis by *J. lividum* were investigated. As shown in Figure 5, both increased oxygen availability and medium dilution significantly enhanced violacein production, with evidence of a synergistic interaction between these two factors.

Violacein production was markedly higher at a k_L_a of 135 h^−1^ than at 34 h^−1^, regardless of dilution. In diluted S3 medium, violacein concentration reached 18.13 mg L^−1^ at a k_L_a of 135 h^−1^, while only 10.07 mg L^−1^ was observed at 34 h^−1^. Similarly, in undiluted S3, production levels were 6.53 and 4.41 mg L^−1^ at 135 and 34 h^−1^, respectively. These results highlight the crucial role of oxygen in violacein biosynthesis, likely due to the involvement of oxygen-dependent enzymes in its metabolic pathway [42].

Furthermore, at both oxygen-transfer rates, diluted S3 culture medium produced consistently higher violacein levels than undiluted S3 medium. This effect was particularly pronounced at the highest k_L_a, achieving an increase in the production of violacein up to 178% (from 6.53 to 18.13 mg L^−1^). Even at lower k_L_a values, an increase of 128% was recorded.

These results suggest that dilution of *scotta* culture medium reduces both nutrient load and culture density, improving oxygen availability in the system. Nutrient-rich media, such as undiluted *scotta*, limit oxygen solubility and facilitate rapid microbial growth, which in turn accelerates oxygen consumption and CO_2_ generation, factors that contribute to hypoxic conditions. Dilution mitigates these effects by moderating growth rates and enhancing oxygen transfer, likely through improved gas–liquid interface dynamics and reduced turbidity [43,44].

The synergistic influences of increasing oxygen transfer and reducing inhibitory conditions via dilution appear essential to the optimization of violacein production in shaking-flask cultures. These observations are consistent with previous findings by Bagoghli and Hosseini-Abari 2024 [9], who reported that violacein yields decreased with an increasing medium-volume-to-air ratio. Specifically, the highest violacein production occurred at 10% medium volume, with yields reduced at 15% and 20%. This supports the idea that, under constant agitation, aeration efficiency is inversely proportional to culture volume in shaking-flasks [45].

#### 3.2.3. Indirect Impact of Lactose Hydrolysis on Pigmentation

Considering that *J. lividum* is a lactose-negative bacterium, the effect of lactose hydrolysis on violacein biosynthesis was evaluated in both diluted and undiluted *scotta* by adding the enzyme lactase.

From the analysis of sugar consumption, see Table 2, key information can be gleaned about how sugar availability affects the ability of *J. lividum* to produce violacein.

Taken together, these data demonstrate a clear synergy between medium dilution and sugar availability in supporting violacein biosynthesis and confirm the results already observed in the previous section, namely, that medium dilution enhances the production of violacein.

As expected, in the S3 sample types (both diluted and undiluted), *J. lividum* DSM1522 was unable to metabolize lactose that remained essentially unconsumed after 120 h, while glucose and galactose were depleted. This is consistent with the Biolog profile of strain DSM1522 and with literature data for several *J. lividum* strains [46,47,48] although, most literature data indicate violacein-producing bacteria belonging to the genus *Janthinumbacterium*, including some strains of *J. lividum*, as lactose-positive bacteria [49].

In S3+L, since lactose hydrolysis was conducted simultaneously with fermentation, glucose and galactose were not detected at time 0 h.

Conversely, in *scotta* S4, at time zero, the values (g L^−1^) of lactose, glucose and galactose were 7.22, 21.86 and 20.36 in the undiluted S4 and 2.74, 8.30 and 7.74 in the diluted S4, respectively.

Unexpectedly, in the undiluted sample types, violacein production was observed only in S3 (6.53 mg L^−1^), while in S3+L and S4 no pigmentation had occurred at 120 h despite glucose and galactose being reduced to 6.49 and 13.39 g L^−1^ in S3+L and to 9.97 and 18.97 g L^−1^ in S4.

This may be attributed to the presence of sugars other than lactose (i.e., glucose and galactose) that accelerate oxygen consumption and CO_2_ generation, factors that reduce oxygen availability in the undiluted medium and may have inhibited violacein biosynthesis, which depends on oxygen-requiring enzymes [42].

Pantanella et al. (2007) [50] claimed that *J. lividum*, in the presence of glucose, can abandon the biofilm state upon entering into planktonic conditions. The authors observed that violacein production was inhibited by the growth of *J. lividum* strain DSM1522 in Luria–Bertani (LB) broth supplemented with 1% glucose.

Similar findings were reported by Fender et al. (2012) [51] for the red pigment prodigiosin, another microbial secondary alkaloid metabolite, which is produced by *Serratia marcescens*. The authors reported that the production of prodigiosin can be repressed by glucose due to acidification of the medium and the decrease in pH.

In contrast, dilution of the *scotta* medium significantly enhanced violacein production. Specifically, in diluted S3, although the lactose remained stable and was not utilized, as in the undiluted case, violacein levels increased to 18.13 mg L^−1^. This result suggests that, even without direct sugar consumption, oxygen transfer is improved due to the lower viscosity of the medium, since k_L_a decreases with increasing liquid viscosity [44].

Additionally, in the diluted *scotta* 3+L condition, complete glucose consumption and significant galactose absorption (13.39→5.27 g L^−1^) were observed, resulting in a violacein yield of 19.87 mg L^−1^.

Finally, the highest violacein concentration (20.21 mg L^−1^) was observed in diluted S4, which had high initial levels of glucose and galactose. Substantial sugar consumption occurred, with glucose dropping from 8.30 to 1.62 g L^−1^ and galactose from 7.74 to 6.95 g L^−1^.

Overall, these data demonstrate a clear synergy between medium dilution and sugar availability in supporting violacein biosynthesis and confirm the results already observed in the previous section, namely, that medium dilution enhances violacein production.

### 3.3. Violacein Production in a Bioreactor at Two Different k_L_a Values

Violacein production was evaluated in a 2 L bioreactor (Figure 6) using diluted *scotta* (S4) derived from lactose-free dairy production as the medium.

The results presented in Figure 7 demonstrate the significant influence of the volumetric oxygen-transfer coefficient (k_L_a) on violacein production, bacterial growth and substrate consumption during fermentation. Specifically, increasing k_L_a from 14.97 h^−1^ to 58.72 h^−1^ resulted in a marked enhancement of violacein production reaching approximately 50 mg L^−1^, compared to just 2.3 mg L^−1^, which was obtained with the lowest k_L_a.

Although the yields obtained in the bioreactor at k_L_a 58.72 h^−1^ were higher than those obtained in flasks at k_L_a 135 h^−1^, it is likely that in the bioreactor, the pH was kept constant at 7.2, while in the flasks, after 120 h, the pH reached values around 9. It has been proved that production in bioreactor gave better results than in stirred-flask studies [15]. These results confirm the findings observed in the flask tests and emphasize the importance of oxygen supply strategies in maximizing the efficiency of *J. lividum* in pigment production in *scotta*-based media.

This observation is in line with findings of Palukurty et al. (2019) [52], which determined that in a bubble column reactor the aeration plays a key role in the production of this secondary metabolite by *Chromobacterium violaceum*.

Glucose and galactose consumption profiles were monitored throughout the fermentation process; the results revealed significant differences in substrate utilization, depending on oxygen availability.

Glucose was rapidly depleted during the early stages of cultivation under both oxygen-transfer conditions. In cultures with a k_L_a of 58.72 h^−1^, glucose was completely consumed within 72 h, whereas in the lower oxygen condition (k_L_a 14.97 h^−1^), residual glucose was still detectable at 96 h and was completely depleted only after 120 h.

Instead, galactose was consumed once the glucose was exhausted. With high k_L_a levels, full consumption was observed after 168 h, while with low k_L_a levels, complete depletion was achieved only after approximately 233 h.

In both conditions, glucose was the preferred carbon source and galactose consumption occurred only after glucose was depleted; this sequential consumption is consistent with a mechanism of carbon catabolite repression [53]. When multiple carbohydrates are present, the mechanism of carbon catabolite repression allows bacteria to preferentially assimilate their preferred carbon source first [54].

Regarding bacterial growth, this is significantly faster with the higher k_L_a (58.72 h^−1^), although both conditions eventually reach similar final biomass levels. This suggests that bacterial biomass accumulation is a key, but not the sole, determinant factor for pigment production Indeed, as reported by Xu et al. (2022), the violacein titers per dry cell weight of biomass (mg g^−1^ DCW) may be different, changing the conditions of fermentation [55].

The results highlight the critical role of oxygen transfer in aerobic fermentations involving *J. lividum* and show a strong influence on violacein formation. A final violacein production of 58.72 mg L^−1^ was observed and, considering the dilution factor (2.70) used for the preparation of the medium, we estimate that from 1 L of *scotta* and using the DSM1522 strain, up to 158.54 mg of violacein can be synthesized.

The violacein yield achieved in this study confirms the average production typically observed in wild-type strains, which generally depends on species and culture conditions [7,50,56].

Although these values remains below the levels reported in processes with genetically engineered hosts such as *Escherichia coli* or *Corynebacterium glutamicum*, these strains have been shown to achieve yields ranging from 1.75 to 5.43 g L^−1^ [57,58]. Process-based optimizations such as L-tryptophan supplementation using agro-industrial substrates can lead to crude violacein yields as high as 1504.5 mg L^−1^ [21].

However, the concentrations reported in these articles are based on the relationships between absorbances and concentrations of crude violacein, without specification of the degree of purity.

Comparing our resulting linear regression equation with those reported in these studies, it is possibly to observe a substantially higher calibration slope. This suggests the higher sensitivity of our assay, which is attributable to the use of a highly purified violacein standard with a purity of >98%.

Furthermore, the consistency of the method was further verified by HPLC determination. Notably, other authors who worked with highly purified violacein standards also reported final violacein concentrations comparable to those observed in our work.

Wu et al. (2021) [59], with the same strain used by us (ATCC 12473), achieved a final maximum violacein concentration of 26 mg L^−1^ when a minimal medium was amended with mannitol.

On the other hand, it has already been suggested that data in the literature regarding violacein yields might be inconsistent and need to be carefully examined [12].

Durán et al. [60] reported that the violacein concentrations presented in many papers are based on the extinction coefficient (ε), the values for which range between 10.9 L g^−1^ cm^−1^ and 74.3 L g^−1^ cm^−1^.

These differences have also been well-highlighted by Rodrigues et al. (2013) [61], who reviewed the analytical approaches utilized for the quantification of violacein. These studies have investigated pure standard solutions in parallel with HPLC and utilizing photometric methods using different extinction coefficients, e.g., ε570 = 10.955 L g^−1^ cm^−1^ [62], ε575 = 29.700 L g^−1^ cm^−1^ [63] and ε575 = 56.010 L g^−1^ cm^−1^ [64], and they highlighted a potential overestimation of the violacein yield of up to 6.8 times higher.

However, in addition to yields, a very recent study [65] highlights that *J. lividum* is used for commercial production of analytical-grade violacein, while the most extensively studied and well-known producer of violacein is *C. violaceum*.

Furthermore, the authors claimed that to overcome the high production costs, and considering the lack of large-scale production methods, the research trend is focusing on optimizing fermentation conditions, using inducers (e.g., L-tryptophan, stressor agents) and using agro-industrial waste as substrates.

Regarding this last possibility, Aruldass et al. (2015) [66] demonstrated the potential application of liquid pineapple waste supplemented with L-tryptophan as an alternative growth medium for the production of violet pigment by *C. violaceum* UTM5, declaring a very high yield (16.257 g L^−1^). In another study, a low-cost soybean-meal-based medium supplemented with L-tryptophan produced approximately 1.504 g L^−1^ of crude violacein [21].

Cassarini et al. (2022) [67] found that selecting protein-rich substrates such as soybean and rapeseed cakes led to improved violacein bioproduction by the addition of tryptophan, using a wild type of *Chromobacterium vaccinii.* Ahmad et al. (2012) [20] achieved pigment production (0.82 g L^−1^) by growing *C. violaceum* (GenBank accession no. HM132057) in sugarcane bagasse supplemented with 10% (*v*/*v*) L-tryptophan.

In this context, *scotta* or whey, especially because they are a good sources of tryptophan [26,68], can be considered as good low-cost substrates for the production of violacein, even using alternative bacteria such as *Chromobacterium* species. Our results, although referring to a strain of *J. lividum*, provide important information regarding this potential use and highlight the importance of maintaining high oxygenation of the medium during cultivation to improve violacein yields.

### 3.4. Spray Drying of Violet Pigment and Its Antibacterial Activity

After obtaining the crude violacein extract, a stabilization process was performed, using a laboratory-scale spray dryer (see Appendix A) to improve solubility and convert the pigment into a storable dry powder.

Spray-drying is a well-established and scalable method to encapsulate hydrophobic (lipophilic) compounds such as violacein, using suitable carriers (e.g., mannitol, starch, maltodextrin, gums and proteins) that facilitate the formation of microcapsules. In the literature, Venil et al. (2015) reported indications regarding the use of this technique for the encapsulation of crude violacein using gum arabic as carrier [37].

In present study, the best operating parameters, as indicated by these authors, were adopted. However, the experiment was performed using mannitol as a carrier instead of gum arabic. Mannitol is a sugar alcohol that presents high chemical stability, is considered compatible with almost all drugs and is widely used in sectors such as the food and pharmaceutical industries [69,70].

A fine wisteria-coloured powder was obtained by feeding the spray dryer at 15 rpm (corresponding to approximately 350 g L^−1^) and maintaining an inlet temperature of 180 °C. The outlet temperature under these operating conditions was approximately 75 °C and some product loss occurred in the walls of the drying chamber. However, the weight of the powder is consistent with that of the mannitol (75 g L^−1^) used for the experiments and a yield of approximately 49.3 ± 0.8 g L^−1^ was achieved. It has been reported that percentage yields with laboratory-scale spray dryers are generally not optimal (20–70%) [71].

During this process, thermal degradation or oxidation could potentially occur, especially with sensitive bioactive molecules [72]. In light of this, a 150 mg spray-dried pigment aliquot was completely solubilized in 1 mL of water, and an agar diffusion assay was performed to verify the stability in terms of antibacterial activity. Different dilutions of encapsuled pigment were tested against *B. subtilis* and after 96 h of incubation, visible inhibition zones were observed on agar plates (Figure 8) for up to 6.05 mg L^−1^ of violacein.

This finding is consistent with previous reports highlighting the susceptibility of Gram-positive bacteria to violacein. Cheng et al. (2022) reported that 130 mg L^−1^ of a crude violacein generated an inhibition zone of 9.6 mm against *B. subtilis* [73]. Notably, recent research has demonstrated that violacein possesses a minimum inhibitory concentration (MIC) of 3.125 mg L^−1^ and a minimum bactericidal concentration (MBC) of 12.5 mg L^−1^ against *Bacillus cereus* [74].

Comparatively, previous studies have reported the activity of violacein against other Gram-positive pathogens, such as *Staphylococcus aureus* and *Staphylococcus epidermidis*, with MICs of 3.9 mg L^−1^ and 20 mg L^−1^, respectively [14,75]. Abedin et al. (2024) reported that soaking a disc in 40 mg L^−1^ violacein demonstrated excellent antibacterial activity against *Streptococcus* sp. and *Listeria monocytogenes* [76]. Interesting data are also reported against plant-pathogenic bacteria; in particular, the bacteriostatic activity of violacein fractions against *Clavibacter michiganensis* VKM Ac-1402 was demonstrated in vitro and in vivo [77].

In general, the antibacterial activity of violacein can be attributed to multiple mechanisms, including disruption of cell membrane integrity [14,78]. Cauz et al. (2019), through combined experiments, demonstrated that the cytoplasmic membrane is the primary target of violacein and showed how violacein severely permeabilizes *B. subtilis* and *S. aureus* cells, with visible discontinuities apparent in the cytoplasmic membrane [79].

The current results therefore strengthen the evidence as to violacein’s potential as a natural antimicrobial compound [80], particularly against Gram-positive bacteria, and demonstrate that the spray-drying process did not affect violacein’s antimicrobial properties. Conversely, they may have improved, as microencapsulated violacein exhibits better solubility than the raw pigment.

Indeed, through a further test, a solubility of 230 mg mL^−1^ in water at room temperature, similar to the values observed for mannitol, was observed [81]; in terms of violacein, this was equivalent to approximately 150 mg mL^−1^.

Since the high hydrophobicity of violacein is a strong limiting factor for its potential therapeutic use, this greater solubility in water makes its application in biological fluids conceivable [82].

The encapsulation of hydrophobic molecules such as violacein in drug delivery systems represents a reasonable alternative, aiming to improve its practical application in the pharmaceutical industry [83].

Therefore, our work represents a pioneering study that highlights the use of mannitol in a carrier trough sprayer drying method, aiming to develop a violacein-based antibacterial agent.

In particular, considering that microencapsulation with mannitol is a strategy used for pulmonary drug delivery through the formation of microspheres that can be inhaled as dry powder [84,85], it is possible to hypothesize the potential formulation of a dry powder inhaler (DPI) containing violacein for the treatment of pulmonary infections arising due to pathogenic bacteria such as *S. aureus* and other Gram-positive bacteria.

However, to achieve this goal, further investigations are needed to characterize the microgranules and obtain all the data required for inclusion in a preclinical development program, such as the evaluations of toxicity and genotoxicity, and a broader in vivo efficacy profile [86].

## 4. Conclusions

This study demonstrated the potential application of *scotta*, a specific dairy waste, as an alternative and economical culture medium for the production of a violet pigment by *J. lividum* DSM1522. Trials conducted in flasks and in a bioreactor highlighted how oxygen transfer strongly affects the formation of violacein in liquid culture, which was significantly enhanced at higher k_L_a values.

Furthermore, it provides interesting information regarding the encapsulation of pigment in mannitol using a laboratory-scale spray dryer.

Specifically, the results demonstrate that this drying process did not affect the antibacterial activity of the violet pigment against *B. subtilis*, a Gram-positive bacterium.

These results are very encouraging for the development of a violacein microcapsule-based antibacterial agent that could be of interest to manufacturers, particularly the pharmaceutical industry.

## Figures and Tables

**Figure 1 microorganisms-13-02125-f001:**
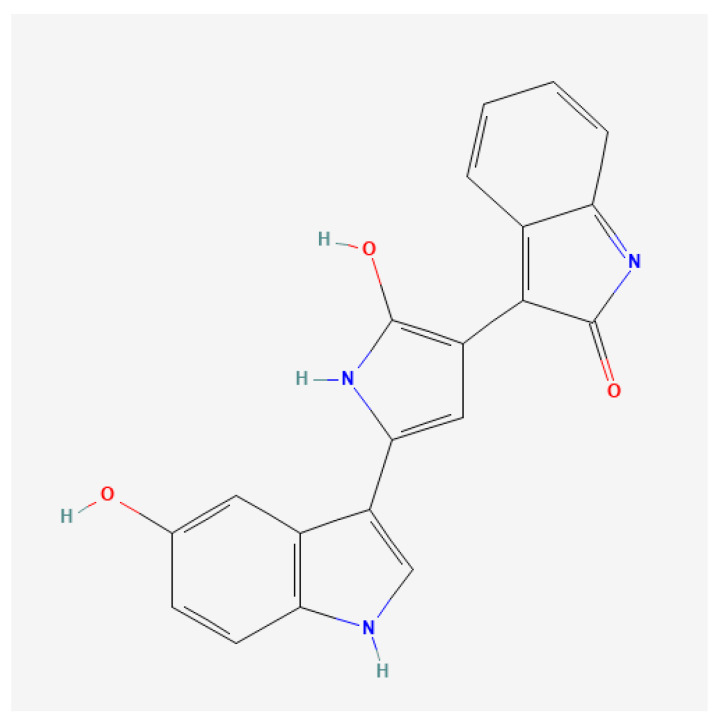
The 2D chemical structure of violacein. Images obtained from PubChem. https://pubchem.ncbi.nlm.nih.gov/compound/11053 (accessed on 2 September 2025).

**Figure 2 microorganisms-13-02125-f002:**
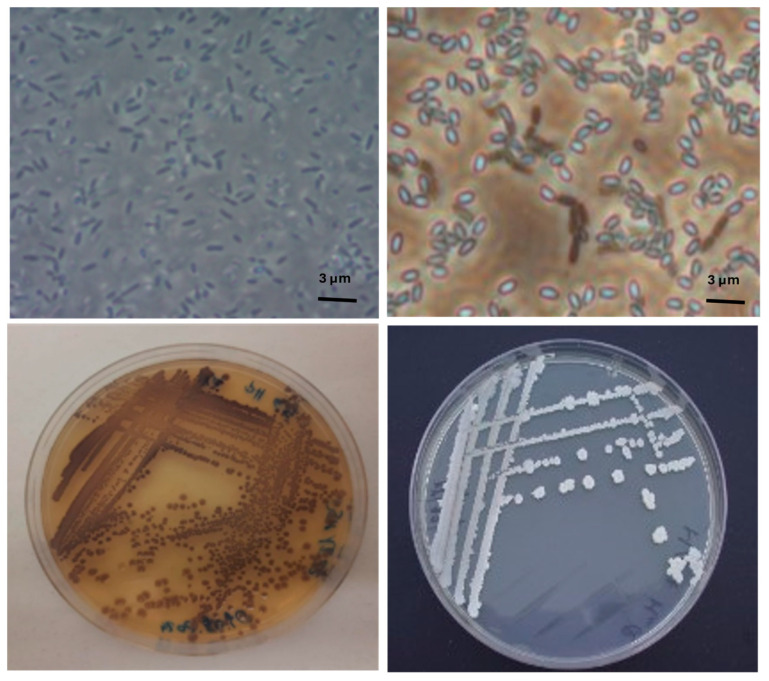
Pure cultures and bacterial cells of *Janthinobacterium lividum* DSM1522 (**left**) and *Bacillus subtilis* ET-1 (**right**).

**Figure 3 microorganisms-13-02125-f003:**
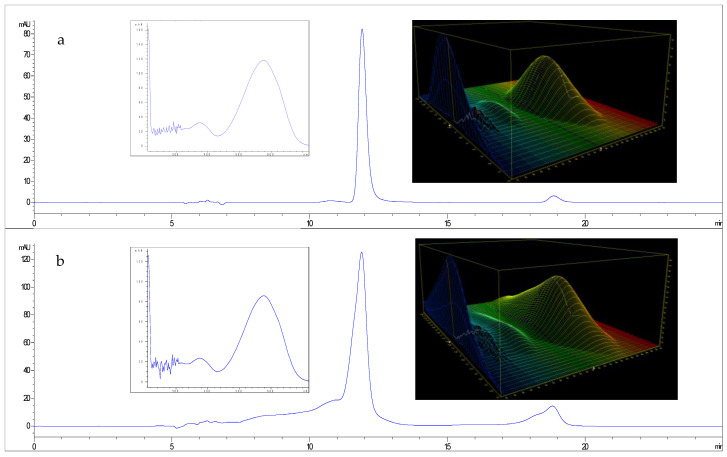
HPLC chromatograms at 580 nm and the relative major peak spectra for violacein extracts obtained from *J. lividum* strain DSM1522 (**a**) and for a standard of violacein at 0.1 g L^−1^ (**b**).

**Figure 4 microorganisms-13-02125-f004:**
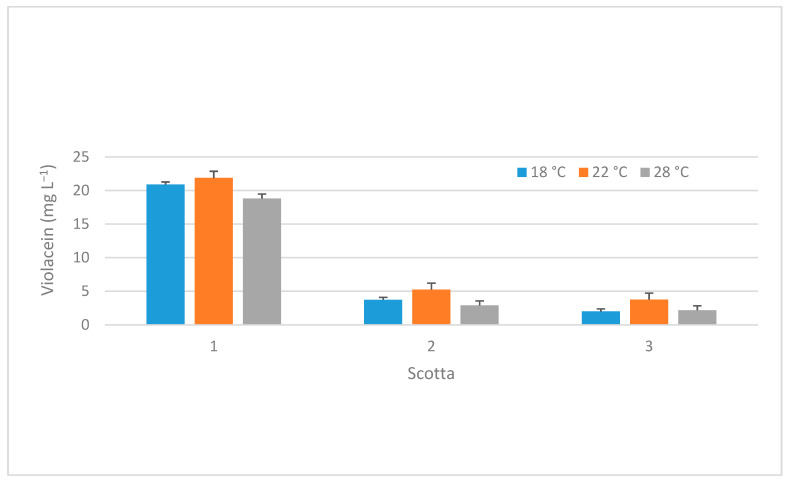
Evaluation of different *scotta* on violacein production at different temperature.

**Figure 5 microorganisms-13-02125-f005:**
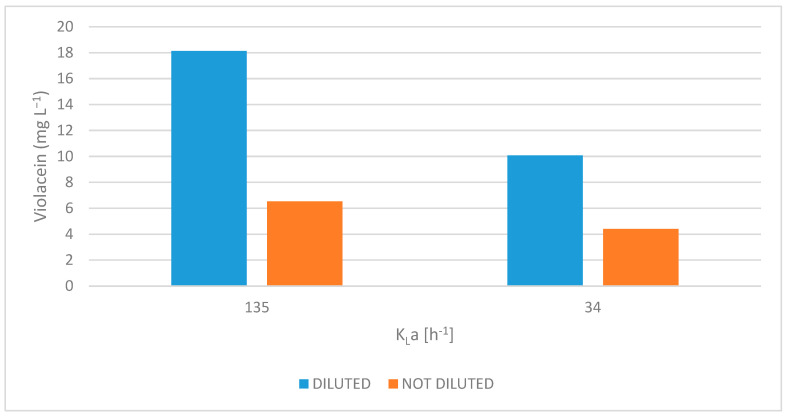
Effects of volumetric mass transfer coefficient (k_L_a) and dilution on violacein production.

**Figure 6 microorganisms-13-02125-f006:**
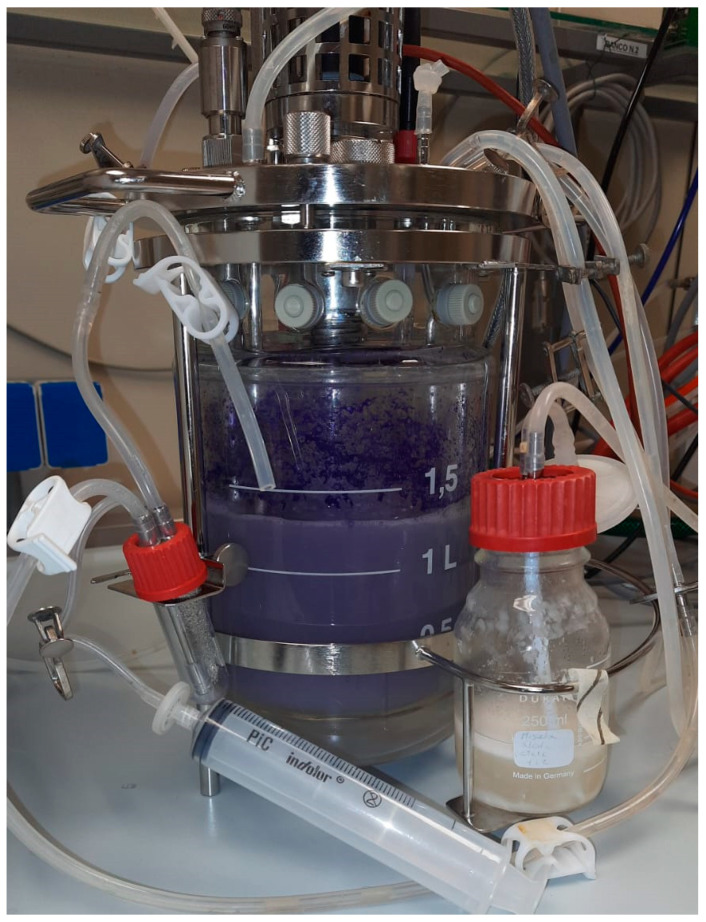
Submerged liquid culture of *J. lividum* strain DSM1522, using *scotta* as medium.

**Figure 7 microorganisms-13-02125-f007:**
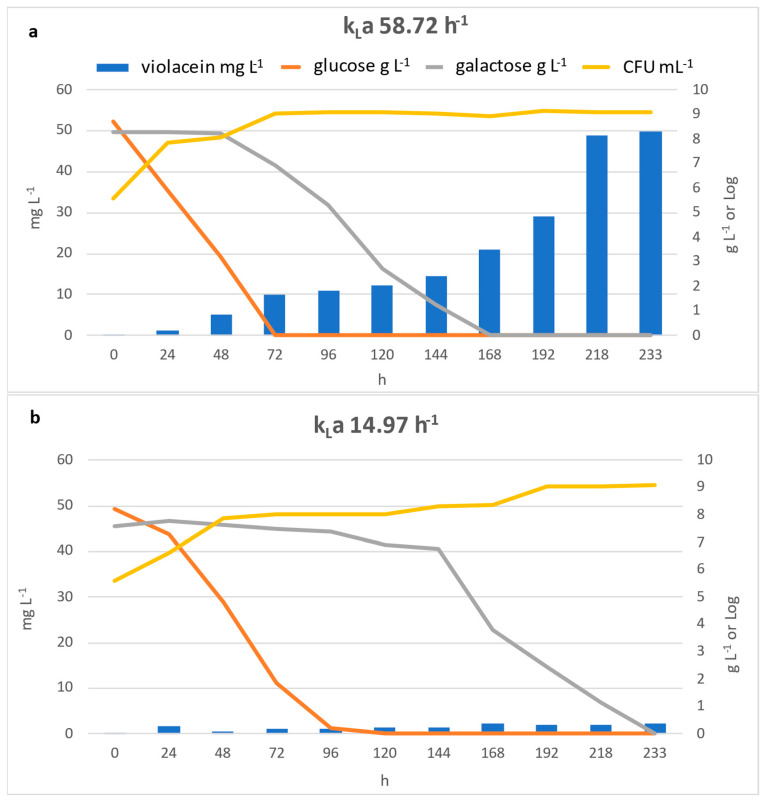
Time course of violacein production, *J. lividum* growth, and sugars utilization in lab-scale bioreactor using diluted *scotta* as medium at different volumetric oxygen-transfer coefficient (k_L_a) values: 58.72 h^−1^ (**a**) and 14.97 h^−1^ (**b**).

**Figure 8 microorganisms-13-02125-f008:**
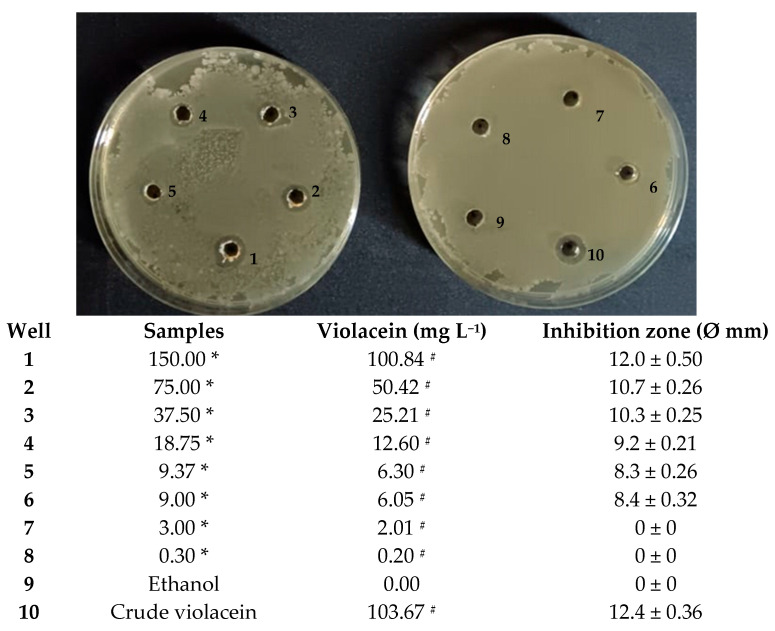
Sensitivity assessment of *B. subtilis* ET-1 relative to spray-dried violacein, using the well-diffusion method. Each well contained 50 µL of spray-dried pigment solubilized in water at different concentrations, ethanol, and crude violacein in ethanol. * Spray-dried pigment solubilized in water (g L^−1^); ^#^ concentration in samples determined by analysis, inhibition zone data are mean values ± standard deviations.

**Table 1 microorganisms-13-02125-t001:** Chemical and physical characterizations of different *scotta*.

*Scotta*	Lactose	pH	Density	EC
	g L^−1^		g mL^−1^	mS cm^−1^
S1	9.99	7.07	1.007	5.15
S2	42.19	6.77	1.015	7.21
S3	26.81	3.9	1.021	19.79
S4	7.22	6.78	1.027	5.89

**Table 2 microorganisms-13-02125-t002:** Violacein production and consumption of sugars by *J. lividum* in flasks containing diluted and not-diluted *scotta*.

Time								
		0 h			120 h			
*Scotta*		Lactose (g L^−1^)	Glucose (g L^−1^)	Galactose (g L^−1^)	Lactose (g L^−1^)	Glucose (g L^−1^)	Galactose (g L^−1^)	Violacein (mg L^−1^)
Not diluted	3	26.81 ± 0.41	ND	ND	26.72 ± 0.32	ND	ND	6.53 ± 0.37
	3+L	26.81 ± 0.41	ND	ND	ND	6.49 ± 0.29	13.39 ± 0.41	ND
	4	7.22 ± 0.34	21.86 ± 0.46	20.36 ± 0.42	6.64 ± 0.11	9.97 ± 0.03	18.97 ± 0.27	ND
Diluted	3	10.19 ± 8.66	ND	ND	10.33 ± 0.41	ND	ND	18.13 ± 0.63
	3+L	10.19 ± 0.34	ND	ND	ND	ND	5.27 ± 0.38	19.87 ± 0.13
	4	2.74 ± 0.12	8.3 ± 0.30	7.74 ± 0.41	2.65 ± 0.03	1.62 ± 0.21	6.95 ± 0.06	20.21 ± 0.44

Data reported are mean value ± standard deviations, ND: not detected.

## Data Availability

The original contributions presented in this study are included in the article/Appendix A. Further inquiries can be directed to the corresponding author.

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
