# Peer review of "Bioconversion of a Dairy By-Product (Scotta) into Mannitol-Stabilized Violacein via Janthinobacterium lividum Fermentation"

_microorganisms, 2025, doi:10.3390/microorganisms13092125_

Round 1
Reviewer 1 Report
Comments and Suggestions for Authors
Manuscript ID: microorganisms-3810283
Title: Bioconversion of a dairy by-product (scotta) into mannitol-stabilized violacein via Janthinobacterium lividum fermentation
Article type: Article
The authors investigated the fermentation of second cheese whey, also named scotta, to produce the organic pigment violacein by the bacterium Janthinobacterium lividum DSM1522.
Firstly, the effect of different types of scotta (including a type derived from the production of lactose-free cheese) on violacein production was studied on a flask scale. The different results obtained by the authors were found to be correlated with differences in lactose concentration, pH, density and conductivity in the by-product samples. Next, the impact of medium dilution and the oxygen transfer coefficient (kLa) was examined. The results demonstrated that both medium dilution and an increased kLa value significantly enhanced violacein production. The authors then studied the fermentation process on a 2 L scale using a bioreactor, as well as the stabilisation of violacein through spray drying using mannitol as a carrier. This resulted in a water-soluble powder that retained antibacterial activity against Bacillus subtilis.
This original article is well developed, clear, and easy to read. The results are very interesting, and the topic is appropriate for the Microorganisms journal. Moreover, the proposed process, from the valorisation of important food waste to the production of an isolated, stable, high-value product, represents a significant advance in knowledge of sustainability and the circular bioeconomy.
I would like to sincerely congratulate the authors on the high quality and potential of this study. On this basis, the manuscript can be accepted for publication following minor revisions.
Minor comments:
- Graphical abstract: Authors should remove the title of their study from the GA.
- Keywords: I suggest replacing the keyword “oxygen transfer” with the term “Jahnthinobacterium lividum”.
- For the second round of revisions, I suggest that authors add line numbers to make revising the manuscript easier.
- Figure 1: Increase the resolution of Figure 1, paying particular attention to the O–H bonds.
- Page 1, line 10: “Chromobacterium violaceum and Janthinobacterium lividum” should remove the italics from “and”.
- Page 1, line 28: revise “byproduct” as “by-product”.
- Page 4, lines 13-14: “However, to the best of our knowledge there are few data regarding the use of these dairy industry by-products as low-cost medium to growth bacteria able to produce violet pigment such as violacein.”. Authors should add references related to the studies containing these data.
- Page 4, line 24-25: “…broth (Peptone 5.0 g L-1; Meat extract 3.0 g L-1)”. Did the authors use a carbohydrate source for the reactivation step? If a carbohydrate source was used, please revise the sentence accordingly.
- Page 5, Figure 2 should be included in the supporting information.
- Page 6, lines 15-16: According to the cited website: “kLa values are based on measurements with PreSens’ SFS v4 plastic shake flasks at 50 mm shaking diameter and PBS buffer in the temperature range 30°C - 37°C. kLa for other diameters can be estimated according to kLa (other) = kLa (50 mm) x (d (other) / 50 mm)33.”. The authors should report in Section 2.3.2 the specific equation adopted in the present study based on the diameter of the flask used in their experiments.
- Page 6, Section 2.3.3 - What is the specific activity of the adopted enzyme (Delact Plus enzyme supplied by Alce International s.r.l.)? What was the rationale behind the authors opting for 1 mL as the volume for the enzyme addition? All these aspects should be clarified in the manuscript.
- Page 7, Section 2.4 - The authors should clarify the selection criteria for the kLa values of 58.72 and 14.97 h⁻¹.
- Page 7, Equation 1 - Place “Equation 1” to the right of the equation and not above it.
- Page 8, line 2: “… and mixed in an orbital shaker for 30 min.”. Authors should add the rpm value and the temperature (room temperature?).
- Page 8, Section 2.6 – “… (approximately 1.0 × 106 CFU mL−1).” Correctly revise the superscripts.
- Page 9, Section 2.8 – “The HPLC chromatograms and UV-Vis spectra were compared (see Error! Reference source not found.) with …”. Revise the sentence.
- Page 10, Figure 3 should be included in the supporting information.
- Page 13, Section 3.2.3 - Why did the authors only carry out tests on S3 and S4 samples? S1 previously produced the best results. The authors should clarify this aspect.
- Page 13, Section 3.2.3 – “From the analysis of sugar consumption, see Table 2Table 2 Violacein production and sugars consumaption by J. lividum in falsks on diluted and not diluted scotta, key information can be gleaned about how sugar availability affects the ability of J. lividum to produce violacein.”. Revise this sentence.
- Page 14, Table 2 - Place the table after it is first mentioned in the text. Avoid using different colours in the same Table and ensure that the tables are consistent in style.
- Page 15, Section 3.3 – “Violacein production was evaluated in a 2-L bioreactor (Figure 6) using diluted scotta (S4) derived from lactose-free dairy production as the medium.”. Why did the authors use sample S4 rather than S1 for testing in the bioreactor? The authors should clarify this aspect in the text.
- Page 15, Section 3.3 - Images should be placed well, and superscripts should be included in the legend along with English terms.
- Page 18, Figure 6 should be included in the supporting information.
- Page 20, revise B. subtilis and Bacillus cereus in italics.
- Page 21, Figure 8 should be included in the supporting information.
Reviewer 2 Report
Comments and Suggestions for Authors
In a manuscript entitled "Bioconversion of a dairy by-product (scotta) into mannitol-stabilized violacein via Janthinobacterium lividum fermentation" Mario Trupo et al. presented results evaluating the potential of using ricotta production waste for the biotechnological production of violacein, a microbial pigment synthesized by the Jantinobacterium lividum strain DSM 1522.
In general, the authors presented quite interesting data on the evaluation of violacein yield depending on the medium compositions and cultivation conditions used for cultivation. The content of the manuscript corresponds to the profile of the journal Microorganisms. However, the manuscript requires some revision, including the design of the presented material.
Unfortunately, the manuscript lacks line numbers, which makes reviewing it somewhat difficult. However, the following positions require revision. In the introduction, the authors have divided the text into short paragraphs, in fact, each sentence represents a separate paragraph, regardless of the semantic load. Please combine sentences into larger paragraphs.
Page 4 of the Introduction. The sentence "Scotta is generally composed of lactose (about 5.0%), salts, short peptides and some residual whey proteins (about 0.2%) which may still contain essential amino acids as tryptophan [33]." is in fact a repetition of the sentence from section 3.1 “In Italy, whey is used to produce ricotta cheese by heating the whey to approximately 90°C. The main by-product of the ricotta production process is scotta, a liquid fraction generally composed of lactose (4.8–5.0%), salts (1.0–1.13%) and proteins (0.15–0.22%)[33][39].” Please modify.
The name of the vioABCDE genes and all bacterial strains should be in italics. Check throughout the text, including the reference list.
Section Materials and Methods. Bacterium Strain or bacterial strain? Also, indicate where the Bacillus subtilis strain was obtained from, what medium was used to cultivate it before analysis and conditions for the test of antimicrobial activity of violacein against this strain.
The data shown in Figure 2: “GEN III microplate biochemical profile (left), pure cultures and bacterial cells of J. lividum DSM1522 (right)” are results of the work, not methods. Consider moving the obtained data to the Results section. In any case, the quality of the presented figure is very low. The legends are illegible. What do the legends in the lower left corner mean? The light microscopy photograph does not have a scale bar. The Materials and Methods do not indicate which microscope was used to image the cells.
Section 2.2 bovine or cow?
Section 2.3.1. The sentence "Before sterilization, the pH was adjusted to approximately 7.0 by adding 1 M NaOH. All flasks were inoculated with 500 μL of suspension obtained by diluting young bacterial colonies" does not allow us to assess how the inoculation was performed. What kind of young bacterial colonies were they? How many hours did they grow? On what medium? How many were there? Strictly speaking, the optical density of the bacterial suspension used for inoculation or the initial OD in the flasks during the experiment is usually indicated.
In the materials and methods in section 2.3.2, conditions are given for 4 variants when studying the effect of mass transfer on the biosynthesis of violacein: two variants of mass transfer (two kLa values of 135 h-1 and 34 h-1) and two variants of flasks, with and without partitions. In the Results section, information on the biosynthesis of violacein in flasks with partitions is missing.
Use the same font to represent the same quantities. For example, on page 7 there are two variants of writing kLa.
Section 2.6. Please, check all superscripts in the formula 1.0 × 106 CFU mL−1. The same for “19.79 mS cm-1” on the page 10.
"Before testing antibacterial activities, 0.15 g of pigments were" in this analysis was 150 mg of pigment or encapsulated sample used? Use the same units to express all quantities. Not 0.15 g, but 150 mg if the concentration is given as mg/ml below.
Figure 3 also contains data on the results obtained, but not materials and methods.
Page 11. The phrase "we can state that S1 produces the highest amount of violacein" requires revision, since, strictly speaking, the S1 medium does not produce violacein; it is synthesized by the bacterium when cultivated in this medium.
Page 12 “This effect was particularly pronounced at higher kLa values, where dilution resulted in a 178% increase in violacein production (from 6.53 to 18.13 mg L⁻¹).”
Throughout the text: unify the spelling of the word "violacein", in some places in the middle of a sentence it is written with a lowercase letter, in other places - with a capital letter. In general, there is no need to write the word violacein with a capital letter in the middle of a sentence.
Page 13. When referring to a table, it is enough to indicate its number, without the title.
Section 3.3 begins with a reference to Figure 6, but the numbering and caption for Figure 6 are missing. Moreover, in the same paragraph (p. 15) a reference is given to Figure 7. Check the numbering. Also check the notations inside the picture and replace Italian with English.
Page 16. It is written: “A final violacein production of 58.72 mg L-1 was observed and, considering the dilution factor (2.70) used for medium preparation, a yield of 158.54 mg of violacein from 1 L of initial scotta can be estimated. The violacein yield achieved in this study (158.54 mg L-1)”.
It seems to me that by using the value obtained as a result of recalculation (58.72 mg/l), the authors distort the real situation. This amount of violacein could be obtained from 2.7 l of diluted medium, but not from a liter of concentrate. Therefore, modify the text, specifying either the actual amount of violacein obtained from 1 liter of diluted medium or indicating the volume of 2.7 l. Expression “158.54 mg L-1” can not be used. Moreover, the authors mislead readers who see the following phrase in the Abstract: "In the bioreactor, a final yield of 158.54 mg of violacein for a liter of scotta was achieved." This does not correspond to the real situation in the fermenter. Please modify.
On page 15 the authors refer to figures 6 and 7. On page 15 there is a figure without a number. On page 18 there is figure 6, on page 19 there is figure 7. Please check the correct placement of materials.
Page 20 - there should be a space between the number and the value (96h), B. subtilis, Bacillus cereus should be italicized.
Page 20. Check whether the following sentence refers to mg/ml or mg/l of violacein: “After 96h of incubation, visible inhibition zones were observed on agar plates (Figure 9) up to the 9.00 mg mL−1 dilution where the corresponding violacein concentration determined by HPLC analyses was approximately 6.05 mg L−1.”
Figure 9. The maximum concentration of encapsulated violacein is 0.15 g/mL (=150 g L-1), as stated in the footnote. How did the authors obtain the violacein concentration of 100.84 mg/L in the same variant? Please check Units for numbers.
Author Response
Please see tha attachment

Round 2
Reviewer 2 Report
Comments and Suggestions for Authors
The authors have made edits to the manuscript. There are no comments on the content. The manuscript can be accepted for publication.